## Invited reply

health and disease and epidemiology/ computational biology

**Author for correspondence:**
Julian Peto
e-mail: julian.peto@lshtm.ac.uk

# Reply to comment on Peto *et al.* (2020): Weekly COVID-19 testing with household quarantine and contact tracing is feasible and would probably end the epidemic

## Julian Peto

London School of Hygiene and Tropical Medicine, London, UK

JP, 0000-0002-1685-8912

An epidemic model is a hypothesis, not an observation. Few scientists would question our statement that the impact of 'the combination of weekly SARS-CoV-2 testing with an earlier test if symptoms appear, strict household quarantine and contact tracing … cannot be reliably predicted by further modelling' [1, p. 3]; yet Planck and colleagues [2] claim that their simple model shows that mass weekly testing and household quarantine, even if it were perfectly achievable, would not be sufficient to control the spread of COVID-19. This is contradicted by the transient reversal of rising prevalence in Slovakia after two rounds of weekly national testing and household quarantine. Prevalence fell by 58% within a week, and a microsimulation calibrated to the observed results confirms that quarantining the whole household following a positive test made a dominant contribution, with an estimated weekly reduction in the prevalence of 59% with and only 10% without household quarantine [3]. We need real data on the effects of different population testing protocols in whole cities [4], not uncalibrated simulations predicting that it cannot work.

The accompanying comment can be viewed at http://doi.org/10.1098/rsos.201546.

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
