## [Peer Review File · Royal Society Open Science]

Review History

RSOS-210467.R0 (Original submission)

Review form: Reviewer 1

Is the manuscript scientifically sound in its present form?

Yes

Are the interpretations and conclusions justified by the results?

Yes

Is the language acceptable?

Yes

Do you have any ethical concerns with this paper?

No

Have you any concerns about statistical analyses in this paper?

No

Recommendation?

Accept as is

Comments to the Author(s)

Overall, I am fine with accepting and publishing this reply as is, but frankly I do not see much of a point to doing so.

I am not too familiar with this publishing model (i.e., a short reply to a previous paper) and I am not a big fan of it either. I understand that the author(s) feel the need to reply. Yet, this reply does not contain any substance (for example, where is the microsimulation?) and I can see that now the other authors will be upset, will want to send a reply etc.

Review form: Reviewer 2

Is the manuscript scientifically sound in its present form?

Yes

Are the interpretations and conclusions justified by the results?

Yes

Is the language acceptable?

Yes

Is it clear how to access all supporting data?

No

Do you have any ethical concerns with this paper?

No

Recommendation?

Accept as is

Comments to the Author(s)

I have no comments

Review form: Reviewer 3

Is the manuscript scientifically sound in its present form?

Yes

Are the interpretations and conclusions justified by the results?

Yes

Is the language acceptable?

Yes

Do you have any ethical concerns with this paper?

No

Have you any concerns about statistical analyses in this paper?

No

Recommendation?

Accept as is

Comments to the Author(s)

I am more inclined to believe the results in the comment by Plank et.al, as the newer evidence points towards a smaller contribution from within-household transmission than Peto et.al. assume. The Reply here seems sensible enough though.

Decision letter (RSOS-210467.R0)

Dear Dr Peto:

I am pleased to inform you that your manuscript entitled "Author reply to comment" is now accepted for publication in Royal Society Open Science.

COVID-19 rapid publication process:

We are taking steps to expedite the publication of research relevant to the pandemic. If you wish, you can opt to have your paper published as soon as it is ready, rather than waiting for it to be published the scheduled Wednesday.

This means your paper will not be included in the weekly media round-up which the Society sends to journalists ahead of publication. However, it will still appear in the COVID-19 Publishing Collection which journalists will be directed to each week (<https://royalsocietypublishing.org/topic/special-collections/novel-coronavirus-outbreak>).

If you wish to have your paper considered for immediate publication, or to discuss further, please notify openscience_proofs@royalsociety.org and press@royalsociety.org when you respond to this email.

on behalf of Professor Mark Chaplain (Subject Editor)

Reviewer(s)' Comments to Author:

Reviewer: 1

Comments to the Author(s)

Overall, I am fine with accepting and publishing this reply as is, but frankly I do not see much of a point to doing so.

I am not too familiar with this publishing model (i.e., a short reply to a previous paper) and I am not a big fan of it either. I understand that the author(s) feel the need to reply. Yet, this reply does not contain any substance (for example, where is the microsimulation?) and I can see that now the other authors will be upset, will want to send a reply etc.

Reviewer: 2

Comments to the Author(s)

I have no comments

Reviewer: 3

Comments to the Author(s)

I am more inclined to believe the results in the comment by Plank et.al, as the newer evidence points towards a smaller contribution from within-household transmission than Peto et.al. assume. The Reply here seems sensible enough though.
